# General Prediction of Interface Chemical Bonding at Metal–Oxide Interface with the Interface Reaction Considered

**DOI:** 10.3390/ma18133096

**Published:** 2025-06-30

**Authors:** Michiko Yoshitake

**Affiliations:** 1National Institute for Material Science, Tsukuba 305-0047, Japan; yoshitake.michiko@nims.go.jp; Tel.: +81-298-63-5496; 2MatQ-lab, Chiba 271-0092, Japan; materials.curation@gmail.com

**Keywords:** metal–oxide interface, interface chemistry, thermodynamic equilibrium, interface reaction, prediction software

## Abstract

A method for generally predicting interface chemical bonding at the metal–oxide interface with the interface reaction considered is reported. So far, the interface between pure metal or alloy and 11 oxides—MgO, Al_2_O_3_, SiO_2_, Cr_2_O_3_, ZnO, Ga_2_O_3_, Y_2_O_3_, ZrO_2_, CdO, La_2_O_3_, and HfO_2_—without considering the interface reaction, has been discussed and implemented in the free web-based software product InterChemBond (v2022). Now, the number of oxides available for prediction is 83 in total. Among them, 29 oxides are in one stable valence, and the others are multi-valence. The newly developed prediction method considering the interface reaction is additionally implemented in InterChemBond. The principles and formula for predicting interface bonding while considering interface reactions are provided as well as some screenshots of the software.

## 1. Introduction

Chemical bonding at the metal–oxide interface is of great interest from both the scientific and practical points of view. Such interfaces are highly important for many practical applications. Strong interface bonds are necessary for solid-state bonding, thermal- or corrosion-resistant coatings, and fabrication of composite materials. Band alignment at the metal–oxide interfaces determines the performance of electric and optical devices, including solar cells. Chemical reactions at metal–oxide interfaces govern the characteristics of catalysts, fuel cells, and batteries. Oxides can have a polar surface wherein the topmost surface is occupied by only oxygen or the metal atoms constituting the oxide. Therefore, the interface with metals can be terminated either by oxygen or metal atoms. Since interface-terminating species have significant influence on the bonding strength, wetting [1,2,3,4,5], and band alignment [6,7,8,9,10,11,12,13], it should be of great importance to develop a method for general prediction to minimize time for material research. For example, dewetting of Au film on glass occurred due to poor bonding strength upon annealing the specimen [14]. So far, we have developed a method to predict interface-terminating species without considering interface reactions, for metal–Al_2_O_3_ [15,16], metal–ZnO [17], and metal–oxide interfaces with nine additional oxides: MgO, SiO_2_, Cr_2_O_3_, Ga_2_O_3_, Y_2_O_3_, ZrO_2_, CdO, La_2_O_3_, and HfO_2_ [18]. Most multi-valence oxides (oxides with different metal valences for the same metal, such as TiO_2_, Ti_2_O_3_, and TiO) were excluded in the previous discussion because such oxides often react with a contacting metal and form oxides with reduced valance. Multi-valence oxides in the previous prediction method were the most stable oxides among multi-valence oxides, and oxides with reduced valence were not considered. The previous prediction method is also applicable to interfaces between elemental semiconductors such as Si and Ge (instead of metals) and various oxides. The prediction method has already been implemented as the free web-based software InterChemBond [19], and anyone can use the software free of charge. From its first release, there have been approximately 200 constant users every year.

When the metal–oxide interface is discussed from the viewpoint of thermal stability at a high temperature or in a catalysis field, the interfacial reaction between metal and oxide matters significantly. Therefore, the author has developed a method for predicting interface chemical bonds with the interface reaction taken into account. In taking the interface reaction between the metal and oxide, oxidation of the metal and reduction of the oxide should be considered. Then, oxides with reduced valence for multi-valence oxides have to be handled appropriately. In this paper, a method is proposed for predicting interface chemical bonds with interface reactions, where the formation of a mixed oxide of A (metal component of the oxide) and M (metal), the oxidation of a metal M (formation of MO), and the formation of alloys between the metal component of oxide A and the contacting metal M, MA are considered based on the thermodynamics between the metal and oxides, including multi-valence oxides. The InterChemBond, in which the newly developed method for prediction with the interface reaction is additionally implemented, is also demonstrated.

## 2. Overview of the Prediction Method Without an Interface Reaction

In the previous method, interface chemical bonding is predicted based on thermal equilibrium at the interface shown in Equation (1):M-A-O − (metal termination) + 1/2O_2_ ⇌ M-O-A-O − (oxygen termination)(1)
where the equilibrium constant K is represented by Equations (2) and (3) by approximating the partial oxygen pressure at the interface to be constant:*K* = exp(−ΔG/RT)(2)G = {chemical potential of (M-O-A-O—) at the standard condition}− {chemical potential of (M-A-O—) at the standard condition}− {half of oxygen chemical potential at the standard condition}(3)

The chemical potential of (M-O-A-O—) (metal termination) is approximated using the M–A bonding energy, and that of (M-A-O—) (oxygen termination) is approximated using the M–O bonding energy. Then, the M–A bonding energy is estimated either from the adsorption energy of A on M (≡X1) or by subtracting the adsorption energy of M on M from that of A on M (≡X2). This subtraction is performed because the values of the adsorption energy include not only the influence of the chemical interaction between A and M but also that of the cohesion energy (upon adsorption, an adsorbed atom becomes part of a solid). The M–O bonding energy is estimated either from the adsorption energy of oxygen on M (O on M, ≡Y1) or by subtracting the dissociation energy of molecular oxygen from the adsorption energy of oxygen on M (≡Y2). The prediction is conducted by comparing the adsorption energy values using the formula below according to the flowchart in Figure 1a:

Approx-1: (A on M) vs. (O on M) (= X1 vs. Y1).

Approx-2: {(A on M) − M on M)} vs. {(O on M) − 1/2(O_2_ dissociation energy)} (= X2 vs. Y2).

When the left side value in Approx-1 or Approx-2 is larger, it suggests that the M-A bond is more stable than the M-O bond. Therefore, an M-A bond is predicted if the left side value is larger in both Approx-1 and -2. On the other hand, the M-O bond is more stable if the right side value in Approx-1 or Approx-2 is larger, resulting in the prediction of a stable M-O bond when the right side value is larger in both Approx-1 and -2. When the side with the larger value differs between Approx-1 and Approx-2, it suggests that the stability of the M-A bond and that of the M-O bond are close. Because Approx-1 and Approx-2 are very simplified approximations of the chemical potential in Equation (3), the values in the formula are expected to deviate from the true chemical potential and can depend on conditions of interface formation, such as temperature and oxygen partial pressure. Therefore, the result of the prediction is “condition dependent”.

The adsorption energy (A on M) is calculated using Miedema’s formula [20] and the software [21] released by the author, which calculates adsorption energies based on Miedema’s formula. The adsorption energy (O on M) is calculated as described in ref. [22], with O_2_ dissociation energy of 493.07 kJ/mol [23]. An example of a Cu-Al_2_O_3_ interface with values for X1, Y1, X2, and Y2 is as follows: the values of X1, Y1, X2, and Y2 are 332, 346.47, 67, and 99.93, respectively. Therefore, in both Approx-1 and Approx-2, the right side value is larger, leading to the prediction of an M-O bond. Although predicting interface bonding using Approx-1 and Approx-2 may seem simplified, the agreement with experimental results exceeds 90%, as discussed in refs. [15,16,17,18]. Therefore, as a tool for screening materials, the prediction method should be useful despite its simplicity, where the influence of crystal orientations, non-stoichiometry, and other factors are all neglected.

For the prediction of interfaces with alloys consisting of metal M_A_ and metal M_B_ (where M_A_ represents the basic metal and M_B_ the additive metal), a chemical equilibrium similar to Equation (1) can be considered. Then, the adsorption energies of A on M_A_ (≡X1), O on M_A_ (≡Y1), A on M_B_ (≡XX1), and O on M_B_ (≡YY1) are used for the chemical potential of different bonds. Here, only the approximation corresponding to Approx-1, used for interfaces with pure metals, is applied to simplify the prediction method (i.e., the influence of cohesive energy is omitted). Just like the comparison between X1 and Y1 in Approx-1 for determining whether the M–A bond or M–O bond is more stable, we compare values among X1, Y1, XX1, and YY1. If XX1 is the largest among the four, this means the adsorption energy of A on M_B_ is the most stable, resulting in the prediction of an A–M_B_ bond at the interface. The prediction of interface bonding is thus performed by identifying which value is the largest among X1, Y1, XX1, and YY1. The flowchart for the prediction is shown in Figure 1b. When X1 is the largest (A on M_A_ is most stable), an A-M_A_ bond is predicted. An O-M_A_ bond is predicted if Y1 is the largest (O on M_A_ is most stable), whereas an O-M_B_ bond is predicted if YY1 is the largest (O on M_B_ is most stable).

It should be noted that, in cases where M_B_ is equal to A (metallic component of oxide), such as an Al-alloy for the metal–Al_2_O_3_ interface and a Zn alloy for the metal–ZnO interface, when O on M_B_ is the largest, the predicted interface is O-M_B_-M_A_(M_B_), which is equal to O-A-M_A_(M_B_), meaning metal-terminated. Detailed discussions on the physical basis of this simplified scheme (compared to the pure metal case) and its limitations in alloyed materials are given in refs. [15,16,17,18].

In Table 1, Table 2 and Table 3 (taken from refs. [15,16,17,18] and modified), both the list of adsorption energy values used for prediction and the comparison between the predicted interface bond and the experimentally observed interface bond are shown for metal (including Al alloy)–Al_2_O_3_ (Table 1), metal (including Zn alloy)–ZnO (Table 2) and metal–several oxides (Table 3) interfaces. It is obvious that the predicted results agree well with experimental results: 12 perfect matches out of 13 combinations in Table 1, where the one disagreement involves both bonds being present due to different experimental conditions; 8 perfect matches out of 10 combinations in Table 2, where one disagreement involves both bonds due to experimental conditions and the other is a disagreement with theory that does not take thermodynamics into account; and 21 perfect matches out of 27 combinations in Table 3, where three disagreements are caused by differences between Approx-1 and Approx-2 predictions, and the other three disagreements involve both bonds being present due to different experimental conditions. Details are discussed in refs. [15,16,17,18]. By comparing X1 vs. Y1 and X2 vs. Y2, one can see that only Rh, Ir, and Pt for Al_2_O_3;_ Pd and Pt for MgO; and Fe for ZrO_2_ are influenced by the type of approximation (whether the effect of cohesive energy is neglected or included). In most cases, one can see that the values of X1 and Y1 are close, with a difference of less than 50 kJ/mol. This issue was also discussed in refs. [15,16,17,18].

Although, in references [15,16,17,18], only a limited number of oxides shown in Table 1, Table 2 and Table 3 are discussed, the equations used for the prediction should be effective for other oxides. Therefore, without considering interface reactions, the above-mentioned prediction method is straightforwardly applicable to 29 single-valence oxides (Li_2_O, BeO, Na_2_O, MgO, Al_2_O_3_, SiO_2_, K_2_O, CaO, Sc_2_O_3_, NiO, ZnO, Ga_2_O_3_, SrO, Y_2_O_3_, ZrO_2_, RuO_2_, Rh_2_O_3_, PdO, CdO, In_2_O_3_, Sb_2_O_3_, Cs_2_O, BaO, La_2_O_3_, HfO_2_, Ta_2_O_5_, OsO_4_, HgO, Bi_2_O_3_). For 54 multi-valence oxides (oxides of 21 elements), the equation used for prediction should be same for the interface between less stable oxides and metals if the interface reaction is not involved. This is because the values of (A on M) and (O on M) are both independent of the valence of the oxides. The 54 multi-valence oxides with 21 elements are TiO_2_, Ti_2_O_5_, Ti_2_O_3_, TiO, V_2_O_5_, V_3_O_5_, V_2_O_3_, VO, CrO_2_, Cr_2_O_3_, Cr_3_O_4_, MnO_2_, Mn_2_O_3_, Mn_3_O_4_, MnO, Fe_2_O_3_, Fe_3_O_4_, FeO, Co_2_O_3_, Co_3_O_4_, CoO, CuO, Cu_2_O, GeO_2_, GeO, As_2_O_5_, As_2_O_3_, RbO_2_, Rb_2_O_2_, Rb_2_O, Nb_2_O_5_, NbO_2_, NbO, MoO_3_, MoO_2_, Tc_2_O_7_, TcO_2_, Ag_2_O_2_, AgO, SnO_2_, SnO, WO_3_, WO_2_, Re_2_O_7_, ReO_3_, IrO_2_, Ir_2_O_3_, PtO_2_, PtO, Tl_2_O_3_, Tl_2_O, PbO_2_, Pb_3_O_4_, PbO. In total, 83 oxides are available for prediction.

## 3. Interface Bonding with Interface Reaction Considered

Now, we consider interface reactions: the formation of a mixed oxide of A and M, the oxidation of metal M (formation of MO), and the formation of alloys between the metal component of the oxide A and the contacting metal M, MA. This general method primarily uses very rough approximations and neglects oxygen partial pressure and the influence of crystal orientations, which means non-stoichiometry is not considered.

For single-valence oxides, it is impossible to reduce the oxide to lower-valence oxides, meaning that the oxidation of metals is impossible. Therefore, the only possible reaction is the formation of mixed oxides.

If a mixed oxide phase exists in the phase diagram, the mixed oxide should always form at the interface. Therefore, the interface will have an M-O-A bond (bonding in mixed oxide), which is regarded as oxygen-terminated. There is no possibility of forming an M-A bond at the interface, regardless of whether the oxide is single-valence or multi-valence. Therefore, the first thing to consider for interface prediction is whether mixed oxides exist or not. As described above, because the formation of a mixed oxide is the only possible interface reaction for single-valence oxides, if no mixed oxide exists for single-valence oxides, the interface bonding prediction is the same as that without interface reaction described in Section 2.

When no mixed oxide phase exists, whether the oxidation of M (M + AO -> MO + AOx (x < 1)) occurs or not is judged using Formula (4) for multi-valence oxides:(Oxide formation enthalpy of metal M per mol-O) − (Oxide formation enthalpy of AO per mol-O)(4)

Here, a value for the oxide with the lowest valence of M should be used for the first term, and a value for the input oxide AO should be used for the second term; both values for various oxides are found in Table 4. If the value of Formula (4) is negative, meaning the oxidation of M is thermodynamically stable, then the interface bond is M – MO – AO, indicating that the interface is predicted to be terminated by O atoms with the formation of MO (oxide of metal M). If the value of Formula (4) is positive, the second formula to judge whether the formation of alloy MA occurs should be calculated, as shown below:
(5)$$\begin{array}{c} \text{Enthalpy of solution of A in M (Mixing enthalpy)}\\ +\;\{(\text{Oxide formation enthalpy of the oxide having a lower valence than the input oxide})\;-\;(\text{That of the input oxide})\}\text{per mol-A atom} \end{array}$$

This formula represents the energy stability of alloy (or intermetallic compound) MA formation by changing the valence of the contacting oxide AO (M + AO -> MA + AOx, where x > 1, meaning that the valence of A in AO increases; for example, Pt + 2SnO -> PtSn + SnO_2_). The first term of Formula (5) is calculated according to ref. [19]. The values of { } in Formula (5) are listed in the fourth column of Table 4. If Formula (5) is negative, the interface is predicted to be terminated with A atoms with the formation of MA. When Formula (5) is positive, it means that no interface reaction occurs: no mixed oxide, no oxide of M (MO), and no alloy of M (MA). The flowchart of the judgement, with images showing the results, is presented in Figure 2. It should be noted that if there are no AOx oxides (x > 1, meaning that the valence of A in AO increases), the formation of MA does not occur.

A step-by-step example for the Fe-TiO interface (Figure 3a) is as follows: regarding Formula (4), the oxide formation enthalpy of Fe per mol-O (= −272.0) minus the oxide formation enthalpy of TiO per mol-O (= −519.7) is positive. The calculation of Formula (5), enthalpy of solution of Ti in Fe (= −86.6) + {(oxide formation enthalpy of the oxide having less valence than the input oxide) − (that of the input oxide)} per mol-A atom (= −240.8), is negative. Therefore, the interface is predicted to be “A-term with MA”, which means Ti-termination with FeTi. This is demonstrated later in Section 4. Another step-by-step example for the Cr-V_2_O_5_ interface (Figure 3b) is as follows: regarding Formula (4), the oxide formation enthalpy of Cr per mol-O (= −382.8) minus the oxide formation enthalpy of V_2_O_5_ per mol-O (= −310.1) is negative. Therefore, the interface is predicted to be “O-term with MO”, which means O-termination with Cr oxide. This is demonstrated later in Section 4. In total, 83 oxides (29 with a single stable valence and the others multi-valence) are included.

As explained above, whether a mixed oxide exists in the involved system should be examined at the beginning of the prediction. If there is a mixed oxide, then the interface is always M-O-A-O bonded, which corresponds to an oxygen-terminated interface with the formation of the mixed oxide. When there is no mixed oxide for the system, whether the oxidation of M occurs should be predicted using Formula (4). If the value of Formula (4) is negative, meaning the oxide of metal M is more stable than AO, the oxidation of metal M and the partial reduction of AO occur. This results in an oxygen-terminated interface with the formation of MO. If there is no mixed oxide and no oxidation of metal M, then the reduction of AO and the formation of an alloy between M and A (MA) should be considered using Formula (5). If the value of Formula (5) is negative, the formation of the alloy MA is stable, resulting in an A-terminated interface with the formation of MA. When there is no mixed oxide, no oxidation of M, and no MA formation, it means no interface reaction occurs, even though the possibility of an interface reaction has been considered. Then, the interface bonding is predicted according to the flowchart of Figure 1a. Therefore, it may happen that the interface bonding without interface reaction is ultimately predicted after considering the interface reactions mentioned above.

For comparison, well-defined experiments—creating interfaces under high vacuum (to ensure well-defined interfaces) and observing interfacial reactions with the valence of oxides examined—are required. To the author’s knowledge, there are only a few experimental results that can be compared with the predictions of the above method, unfortunately. However, Cr_2_O_3_ formation at the interface between Cr and Ga_2_O_3_ [24] and Ga_2_O_3_ formation at the GaSe/SnO_2_ interface [25] have been reported; these correspond to O-term with Cr-O bond and O-term with Ga-O bond as predicted by the above method, respectively. For the interface between Pd and FeOx, the formation of Fe_3_O_4_ and Pd^2+^ has been reported [26], while O-term with Pd-O bond (where Pd is ionized) is predicted for the interface between Pd and Fe_2_O_3_. At the interface between W and CuO, the presence of metallic Cu and WOx has been reported [27], which coincides with the prediction of O-term with W-O bond.

## 4. InterChemBond: Prediction Software

In this section, the free software InterChemBond, which predicts and displays results based on the method described in this manuscript, is explained. A screenshot of the first page of InterChemBond is shown in Figure 4. There are basically three models: (1) interface with pure metal without considering interface reaction, (2) interface with pure metal including interface reaction, and (3) interface with alloy without considering interface reaction. Models 1 and 3 are explained briefly in Section 2; their details have already been published [15,16,17,18].

When “Interface with pure metal including interface reaction” (model 2) on the first page is chosen, an input screen of metal M and oxide AO appears. When a metal (including an elemental semiconductor) and an oxide are chosen for the periodic tables, an inquiry window appears asking whether a mixed oxide phase exists between the selected metal and oxide. A user should choose yes or no. This is because the developer of this system cannot obtain the complete set of phase diagrams for all combinations of metal and oxide and would like the user to provide input according to their knowledge.

In the following, some examples of model 2 in InterChemBond are demonstrated. Examples of interfaces forming mixed oxides are shown for both single-valence oxide (Fe/Al_2_O_3_: Figure 5a) and multi-valence oxide—for a high, stable valence state (Fe/TiO_2_: Figure 5b) and for a low-valence state of the corresponding multi-valence oxide (Fe/TiO: Figure 5c). In the phase diagram of the Fe-Al-O system, a mixed oxide phase, FeAl_2_O_4_, exists [28]. This mixed oxide has actually been observed at the Fe/Al_2_O_3_ interface [29]. For the Fe-Ti-O system, mixed oxides such as TiFeO_3_, TiFe_2_O_5_, and TiFe_2_O_4_ exist depending on temperature [30,31]. As explained in Section 3, if a mixed oxide exists—whether the oxide is single-valence or multi-valence, or whether it has a stable valence state or low-valence state of a multi-valence oxide—the interface is oxygen-terminated with the formation of the mixed oxide.

Examples of cases where no mixed oxide is formed are provided in Figure 6 and Figure 7. Figure 6 demonstrates the different prediction results for different valence states of the same metal and multi-valence oxide (Fe/TiO and Fe/TiO_2_). When low-valence TiO contacts Fe and no mixed oxide formation is assumed, the reaction Fe + TiO -> FeTi + TiO_1+x_ at the interface is expected based on the corresponding enthalpy values. Then, the interface is predicted to be “A-term with MA”, which means Ti-terminated, i.e., an Fe-Ti bond at the interface with the formation of Fe-Ti alloy or intermetallic compound. In the case of contact with high-valence TiO_2_ and no mixed oxide formation assumed, the interface reaction Fe + TiO_2_ -> FeO_x_ + TiO_2−x_ is expected, giving the interface prediction of “O-M bond”, meaning no interface reaction at the interface. The oxidation of Fe and reduction of TiO_2_ were experimentally observed at the Fe/TiO_2_ interface [32].

When comparing Figure 6b with Figure 5b, both concerning the interface between Fe and TiO_2_, the predictions differ because the user’s choice on the formation of mixed oxide was different. The software simply follows the user’s choice, whether that choice is correct or not. Depending on the process of interface formation, such as low-temperature treatment, a mixed oxide may not form even if it exists in the phase diagram.

In Figure 7, another example of different predictions for different valence states of the same metal and multi-valence oxide (Cr/VO and Cr/V_2_O_5_) is demonstrated. When low-valence VO contacts Cr and no mixed oxide formation is assumed, the interface is predicted to be “A-term with MA”, which means V-terminated, i.e., a Cr-V bond at the interface with the formation of a Cr-V alloy or intermetallic compound. In the case of contact with high-valence V_2_O_5_ and no mixed oxide formation assumed, the interface is predicted to be “O-term with MO”, meaning an O-Cr interface bond and the formation of Cr oxide at the interface. To the author’s knowledge, there are no experimental results on a mixed oxide at the interface, but Cr[VO_4_] exists in the phase diagram of the Cr-V-O system [33]. It should be noted that, for the cases in Figure 6 and Figure 7 (Fe/TiO vs. Fe/TiO_2_ and Cr/VO vs. Cr/V_2_O_5_), by choosing the formation of mixed oxide, the prediction for all the interfaces is “O-term with mixed oxide”.

## 5. Conclusions

In addition to our previous works—the prediction of the interface between pure metal or alloy and 11 oxides (MgO, Al_2_O_3_, SiO_2_, Cr_2_O_3_, ZnO, Ga_2_O_3_, Y_2_O_3_, ZrO_2_, CdO, La_2_O_3_, and HfO_2_) without considering interface reaction—predictions of interfaces with elemental semiconductors instead of metals, and many other oxides (83 in total, including 29 oxides with single stable valence and the others multi-valence), have been included using the previous method. A new prediction method has been developed to account for the influence of interface reactions on interface bonding. The principles and formulas for predicting interface bonding while considering interface reactions are explained. The oxide formation enthalpy of the contacting oxide AO per mol-O and the difference in oxide formation enthalpy between lower-valence and higher-valence oxides, in per mol-A-atom units, are key for the prediction.

Values of the adsorption energy and various oxide formation enthalpies used for the prediction and the flowchart for prediction are implemented in the software InterChemBond, which is free to use. Examples of typical prediction results are demonstrated using screenshots of InterChemBond. In the flowchart, the existence of a mixed oxide for the system of interest should be considered at the beginning; this is not implemented as an automatic decision in the software, and users should decide based on the conditions of their experiments.

Although the prediction method uses simple approximations and is very rough, it can be applied to many combinations of metal–oxide interfaces in general. The number of combinations—50 metals and {(50∙49)/2} alloys (combinations of a base metal and additional metal) and 83 oxides—is 105,825 in total. Therefore, the software should be helpful for screening materials. Since there are a very limited number of well-defined experiments that can be compared with the predicted results of this method, further validation using more well-defined experimental results is preferable.

It should be noted that the method can be applied to heterophase interfaces such as metal–oxide–metal, if each interface is independent (i.e., interface reactions at one interface do not affect the other interfaces).

## Figures and Tables

**Figure 1 materials-18-03096-f001:**
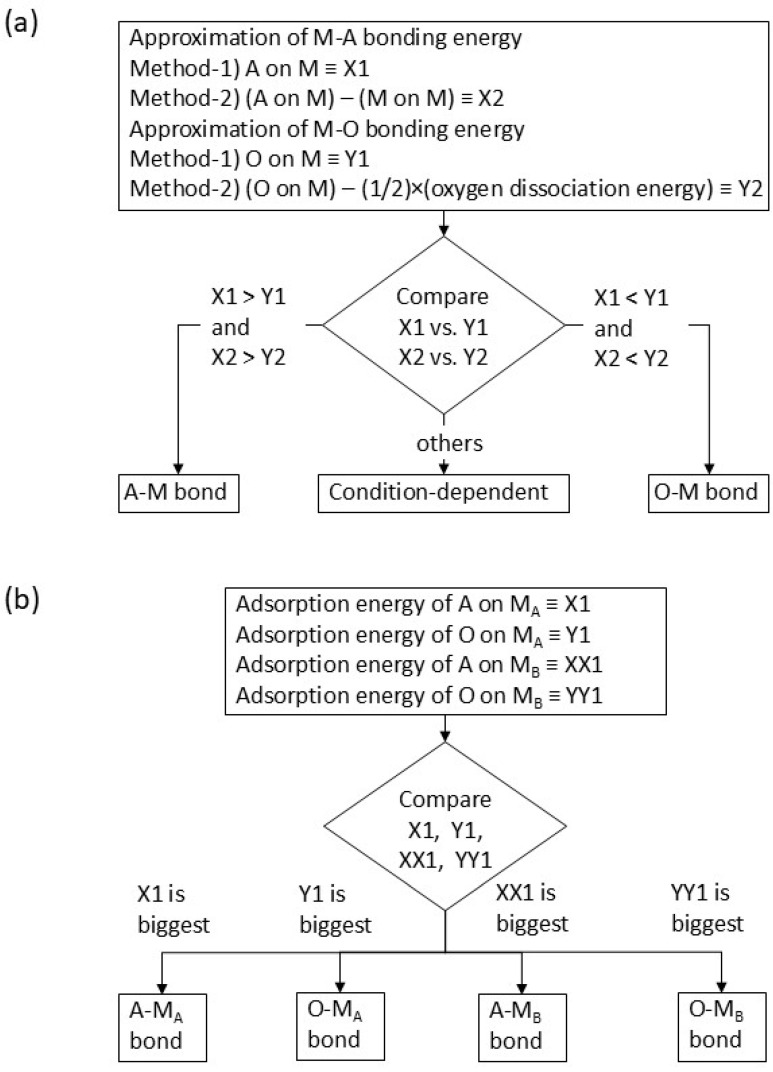
Flowchart for prediction interface bonding for (**a**) pure metal–oxide and (**b**) alloy–oxide interfaces.

**Figure 2 materials-18-03096-f002:**
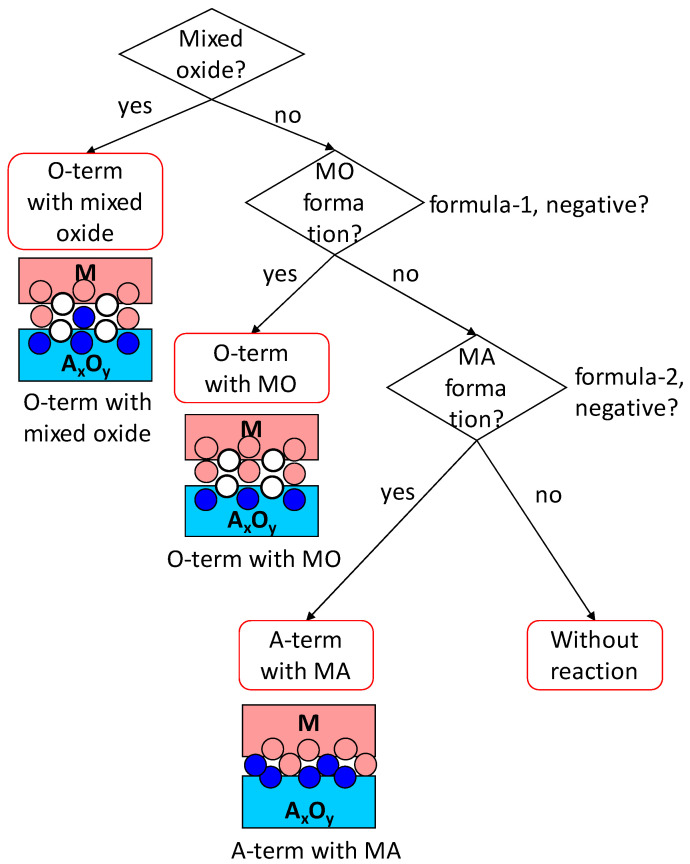
Flowchart for prediction of interface bonding with the interface reaction considered. White, blue, and pink circles are schematic representations of oxygen, metal component A of oxide, and metal M atoms, respectively.

**Figure 3 materials-18-03096-f003:**
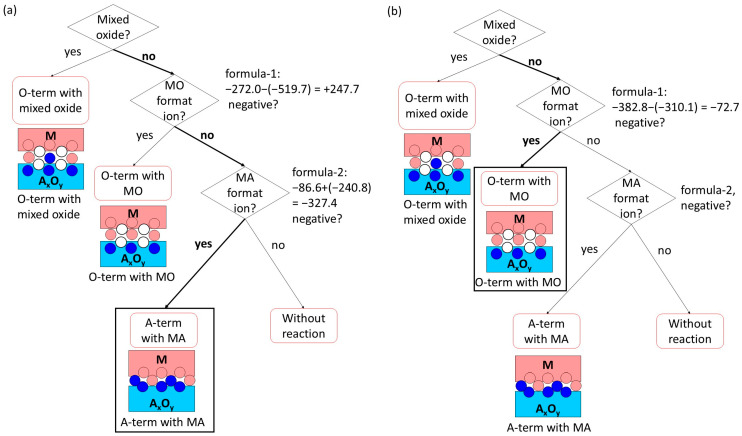
Examples of flowchart application for Fe-TiO (**a**) and Cr-V_2_O_5_ (**b**) systems with the choice of no mixed oxide. White, blue, and pink circles are schematic representations of oxygen, metal component A of oxide, and metal M atoms, respectively.

**Figure 4 materials-18-03096-f004:**
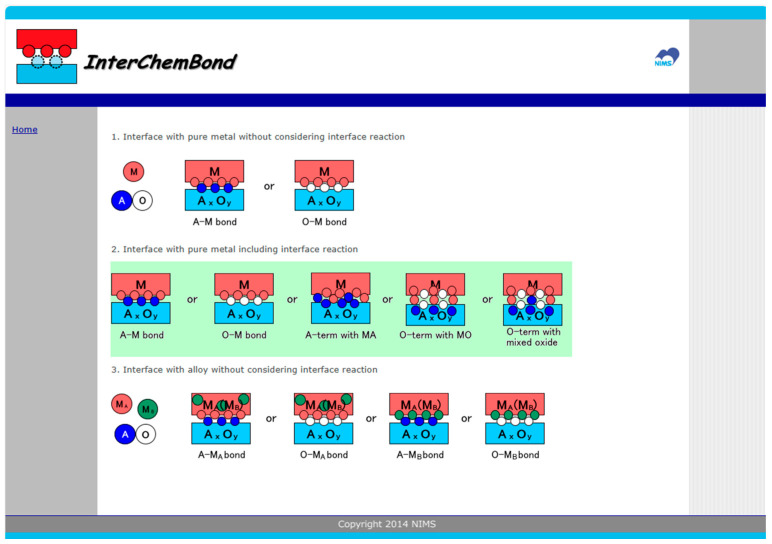
A screenshot of the first page of the current InterChemBond system, which gives predictions of interface bonding for various metal–oxide interfaces with and without the consideration of interface reactions.

**Figure 5 materials-18-03096-f005:**
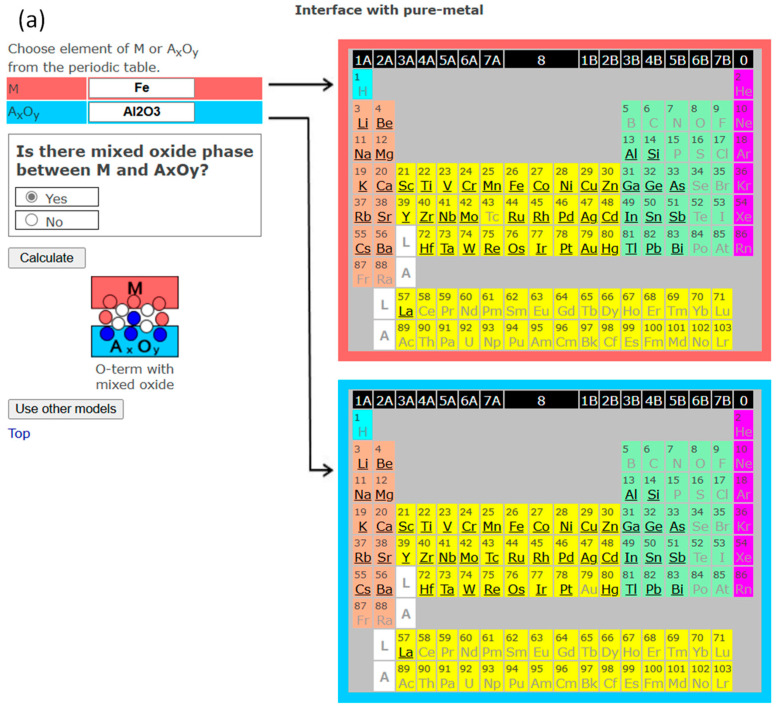
(**a**). The screenshot of InterChemBond system for the interface making mixed oxides for Fe/Al_2_O_3_. (**b**). The screenshot of InterChemBond system for the interface making mixed oxides for Fe/TiO_2_. (**c**). The screenshot of InterChemBond system for the interface making mixed oxides for Fe/TiO. White, blue, and pink circles are schematic representations of oxygen, metal component A of oxide, and metal M atoms, respectively.

**Figure 6 materials-18-03096-f006:**
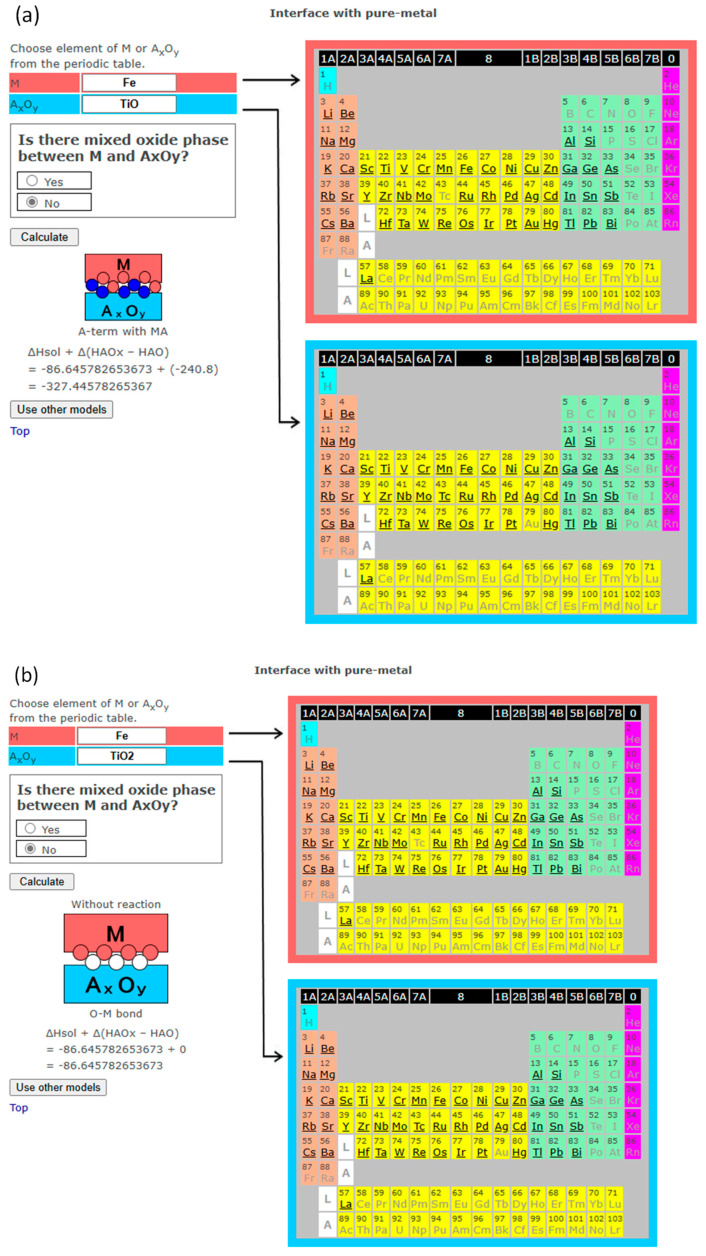
(**a**). The screenshot of InterChemBond system predicting the interface bonding for Fe/TiO system without mixed oxide at the interface. (**b**). The screenshot of InterChemBond system predicting the interface bonding for Fe/TiO_2_ system without mixed oxide at the interface. White and blue circles are schematic representations of oxygen and metal A atoms, respectively.

**Figure 7 materials-18-03096-f007:**
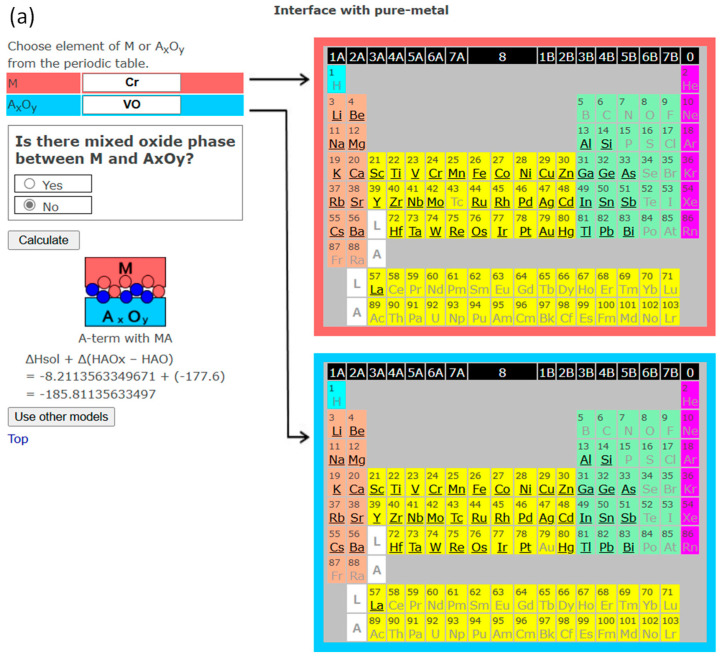
(**a**). The screenshot of InterChemBond system predicting the interface bonding for Cr/VO system without mixed oxide at the interface. (**b**). The screenshot of InterChemBond system predicting the interface bonding for Cr/V_2_O_5_ system without mixed oxide at the interface. White, blue, and pink circles are schematic representations of oxygen, metal component A of oxide, and metal M atoms, respectively.

**Table 1 materials-18-03096-t001:** The list of adsorption energies for the prediction of metal M/Al_2_O_3_ or alloy M(MB)/Al_2_O_3_ interface bonds with predicted results and experimental results. References for experimental results are given in refs. [15,16]. “<0” in the table means negative adsorption energy where the adsorption is energetically unfavorable. “-” shows that values are not necessary for the predictions and not given (only for alloys).

M		Al on M [kJ/mol-M]	(Al on M) − (M on M) [kJ/mol-M]	O on M [kJ/mol-O]		Al on Al	O on Al	Predicted Termination	Experimental Results from Reference
	MB	X1	X2	Y1	Y2	XX1	YY1		
Ti		384	21	908.96	662.43	-	-	O	O
Ti	Al	384	21	908.96	662.43	270	833.06	O	Al, O
V		400	−1	788.34	541.8	-	-	O	O
Cr		377	74	641.44	394.91	-	-	O	O
Fe		392	76	528.65	282.12	-	-	O	O
Fe	Al	392	76	528.65	282.12	270	833.06	Al	Al
Co		408	73	446.36	199.82	-	-	O	O
Ni		407	67	409	162.46	-	-	O	O
Ni	Al	407	67	409	162.46	270	833.06	Al	Al
Cu		332	67	346.47	99.93	-	-	O	O
Cu	Al	332	67	346.47	99.93	270	833.06	Al	Al
Nb		409	−173	913.07	666.54	-	-	O	O
Rh		447	12	356.62	110.09	-	-	Al, O	
Pd		413	130	295.06	48.53	-	-	Al	
Ag		280	58	242.69	−3.85	-	-	Al	Al
Ir		474	−61	429.98	183.45	-	-	Al, O	
Pt		450	2	329.31	82.77	-	-	Al, O	
Au		325	32	<0	<0	-	-	Al	

**Table 2 materials-18-03096-t002:** The list of adsorption energies for the prediction of metal M/ZnO or alloy M(MB)/ZnO interface bonds with predicted results and experimental results. References for experimental results are given in ref. [17]. “<0” in the table means negative adsorption energy where the adsorption is energetically unfavorable. “-” shows that values are not necessary for the predictions and not given (only for alloys).

M		Zn on M [kJ/mol-M]		O on M [kJ/mol-O]		A on M_B_	O on M_B_	Predicted Termination	Experimental Results from References	Theory from Reference
	MB	X1	X2	Y1	Y2	XX1	YY1			
Mg		81	−17	664.14	417.61	-	-	O		O
Ti		184	−179	908.96	662.43	-	-	O	O	
V		198	−203	788.34	541.80	-	-	O		
Cr		177	−126	641.44	394.91	-	-	O	O	
Fe		194	−122	528.65	282.12	-	-	O	O	
Co		212	−123	446.36	199.82	-	-	O		
Ni		211	−129	409.00	162.46	-	-	O		
Cu		152	−113	346.47	99.93	-	-	O	O	O
Cu	Zn	152	−113	346.47	99.93	113	484.61	Zn	Zn, O	
Ge		85	−212	648.70	402.17	-	-	O	O	
Nb		205	−377	913.07	666.54	-	-	O		
Pd		221	−62	295.06	48.53	-	-	O		Zn
Ag		104	−118	242.69	−3.85	-	-	O		
Ag	Zn	104	−118	242.69	−3.85	113	484.61		Zn	
Pt		256	−192	329.31	82.77	-	-	O	O	
Au		144	−149	<0	<0	-	-	Zn	Zn	Zn

**Table 3 materials-18-03096-t003:** The list of adsorption energies for the prediction of the interface bonds between metal M or alloy M_A_(M_B_) and various oxide AO with predicted results and experimental/theoretical results. References for experimental/theoretical results are given in ref. [18]. <0–493.07/2 means that values are the results of negative value minus (493.07 divided by 2), resulting in less than −246.5. “-” shows that values are not necessary for the predictions and not given (only for alloys).

Oxide	Metal-A	Metal-B	Formation Enthalpy of Oxide [kJ/mol]	Formation Enthalpy of Oxide [kJ/mol-A]	Adsorption Energy [kJ/mol]						Prediction	Experiment	Theory
AO	M_A_	M_B_			AonM_A_	OonM_A_	AonM_A_-M_A_onM_A_	OonM_A_-493.07/2	AonM_B_	OonM_B_			
					X1	Y1	X2	Y2	XX1	YY1			
MgO	Cu		601.6	601.6	223	346.47	−42	99.935	-	-	O	O	O
	Cu	Ag			223	346.47	−42	99.935	160	242.69	O-Cu	O	
	Pd				324	295.06	41	48.525	-	-	Mg, O	O	O
	Ag				160	242.69	−62	−3.845	-	-	O	O	O
	Co				291	446.36	−44	199.825	-	-	O		O
	Fe				257	528.65	−59	282.115	-	-	O		O
	Ni				295	409	−45	162.465	-	-	O		O
	Pt				376	329.31	−72	82.775	-	-	Mg, O		O
	W				307	836.67	−388	590.135	-	-	O		O
SiO_2_	Al		910.7	910.7	359	833.06	89	586.525	-	-	O	O	
	Au				395	<0	102	<0	-	-	Si	Si	
Cr_2_O_3_	Ni		1139.7	569.85	313	409	−27	162.465	-	-	O		O; O
Ga_2_O_3_	Cr		1089.1	544.55	363	641.44	136	394.905	-	-	O	O	
Y_2_O_3_	Ge		1905.3	952.65	389	648.7	92	402.165	-	-	O	O	
ZrO_2_	Si		*1094.3*	*1094.32*	476	885.15	117	638.615			O		O
	Fe				588	528.65	272	282.115	-	-	Zr, O		O
	Co				622	446.36	287	199.825	-	-	Zr	O	
	Ni				629	409	289	162.465	-	-	Zr	O; Zr	Zr, O; O
	Cu				529	346.47	264	99.935	-	-	Zr	O	O
	Pd				660	295.06	377	48.525	-	-	Zr	Zr	
	Au				566	<0	273	<0–493.07/2			Zr	Zr	
CdO	Ag		258.4	258.4	112	242.69	−110	−3.845	-	-	O	O	O
	Ag	Au			112	242.69	−110	−3.845	159	<0	O-Ag	Au-seg	
La_2_O_3_	Ge		1793.7	896.85	435	648.7	138	402.165	-	-	O		O
	Si				459	885.15	100	638.615	-	-	O	O; O	
HfO_2_	Si		1144.7	1144.7	444	885.15	85	638.615	-	-	O	O, Hf	O
	Si				444	885.15	85	638.615	-	-	O	Hf, O	
	Pt				671	329.31	223	82.775			O	Hf, O	

**Table 4 materials-18-03096-t004:** The list of formation enthalpy of oxides and the values of { } in Formula (5) (the fourth column) for various oxides with various valences.

Metal Component of Oxide	Oxide	Formation Enthalpy [kJ/mol-O]	Difference of Formation Enthalpy Between AO and A_1−x_O [kJ/mol-A]	Metal Component of Oxide	Oxide	Formation Enthalpy [kJ/mol-O]	Difference of Formation Enthalpy Between AO and A_1−x_O [kJ/mol-A]
Li	Li_2_O	−597.9	-	Sr	SrO	−592.0	-
Be	BeO	−580.1	-	Y	Y_2_O_3_	−635.1	-
Na	Na_2_O	−414.2	-	Zr	ZrO_2_	−550.3	-
Mg	MgO	−601.6	-	Nb	Nb_2_O_5_	−379.9	-
Al	Al_2_O_3_	−558.6	-		NbO_2_	−398.1	−153.6
Si	SiO_2_	−455.4	-		NbO	−405.8	−390.4
K	K_2_O	−361.5	-	Mo	MoO_3_	−248.4	-
Ca	CaO	−634.9	-		MoO_2_	−294.5	−156.2
Sc	Sc_2_O_3_	−636.3	-	Tc	Tc_2_O_7_	−159.1	-
Ti	TiO_2_	−472.0	-		TcO_2_	−228.9	−99.1
	Ti_3_O_5_	−491.9	−124.2	Ru	RuO_2_	−152.5	-
	Ti_2_O_3_	−507.0	−59.4	Rh	Rh_2_O_3_	−114.3	-
	TiO	−519.7	−240.8	Pd	PdO	−85.4	-
V	V_2_O_5_	−310.1	-	Ag	Ag_2_O_2_	−12.2	-
	V_3_O_5_	−386.6	−131.0		Ag_2_O	−31.1	3.4
	V_2_O_3_	−406.3	−34.9	Cd	CdO	−258.4	-
	VO	−431.8	−177.6	In	In_2_O_3_	−308.6	-
Cr	CrO_2_	−299.0	-	Sn	SnO_2_	−288.8	-
	Cr_2_O_3_	−379.9	−28.2		SnO	−280.7	−296.9
	Cr_3_O_4_	−382.8	−59.5	Sb	Sb_2_O_3_	−240.1	-
Mn	MnO_2_	−260.0	-	Cs	Cs_2_O	−345.8	-
	Mn_2_O_3_	−319.7	−40.5	Ba	BaO	−548.0	-
	Mn_3_O_4_	−347.0	−16.9	La	La_2_O_3_	−597.9	-
	MnO	−385.2	−77.4	Hf	HfO_2_	−572.4	-
Fe	Fe_2_O_3_	−274.7	-	Ta	Ta_2_O_5_	−409.2	-
	Fe_3_O_4_	−279.6	−39.3	W	WO_3_	−281.0	-
	FeO	−272.0	−100.8		WO_2_	−294.9	−253.2
Co	Co_2_O_3_	−80.8	-	Re	Re_2_O_7_	−177.2	-
	Co_3_O_4_	−222.8	175.9		ReO_3_	−204.8	−5.8
	CoO	−237.9	−59.1	Os	OsO_4_	−98.5	-
Ni	NiO	−240.1	-	Ir	IrO_2_	−137.1	-
Cu	CuO	−157.3	-		Ir_2_O_3_	−67.5	−173.0
	Cu_2_O	−168.6	−73.0	Pt	PtO_2_	−66.7	-
Zn	ZnO	−350.5	-		PtO	−70.0	−63.4
Ga	Ga_2_O_3_	−363.0	-	Au		0.0	-
Ge	GeO_2_	−290.0	-	Hg	HgO	−90.8	-
	GeO	−261.9	−318.1	Tl	Tl_2_O_3_	−131.0	-
As	As_2_O_5_	−185.0	-		Tl_2_O	−178.7	−107.2
	As_2_O_3_	−444.9	204.9	Pb	PbO_2_	−138.7	-
Rb	RbO_2_	−139.4	-		Pb_3_O_4_	−179.6	−37.9
	Rb_2_O_2_	−236.0	−42.8		PbO	−219.0	−20.5
	Rb_2_O	−339.0	−66.5	Bi	Bi_2_O_3_	−191.3	-

## Data Availability

The original contributions presented in this study are included in the article. Further inquiries can be directed to the author.

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
