# Peer review of "General Prediction of Interface Chemical Bonding at Metal–Oxide Interface with the Interface Reaction Considered"

_materials, 2025, doi:10.3390/ma18133096_

Round 1
Reviewer 1 Report
Comments and Suggestions for Authors
This manuscript introduces an expanded predictive framework for determining the chemical bonding nature at metal–oxide interfaces, incorporating the thermodynamic implications of interface reactions. Building on the author’s prior model implemented in the free software InterChemBond, this work significantly extends the capability from 11 oxides to 83, including multi-valence oxides, and introduces reactions such as oxide reduction, metal oxidation, and mixed oxide formation. Using simplified thermodynamic approximations, the model predicts whether bonding is oxygen- or metal-terminated.
1.The current use of simplified approximations (X1, X2, Y1, Y2) lacks rigorous justification or quantified error margins.
2.Only a few examples are cited to validate the model predictions against experimental data.
3.The selection of only Approx-1 logic for alloy systems oversimplifies potential contributions of cohesive energy differences.4.Users must manually decide if a mixed oxide exists, potentially leading to inconsistent inputs. Integrate or reference a publicly accessible phase diagram database (e.g., CALPHAD-based) to support more objective input.5.The manuscript lacks detailed explanation of the user interface and input/output parameters of the InterChemBond software.
6 There are several grammatical inconsistencies and awkward phrasings (e.g., "dump screen" should be "screenshot").
Reviewer 2 Report
Comments and Suggestions for Authors
The work is devoted to the development of a universal method for predicting chemical bonding at the metal-oxide interface taking into account possible interface reactions, which is a significant step forward compared to previously existing approaches. The problem of forming strong and stable interfaces in metal-oxide systems is of critical importance for a wide range of technologies - from heat-resistant coatings to electronic and optoelectronic devices. Considering that the interaction of metals with oxides can be accompanied by complex chemical transformations, including the formation of mixed oxides, oxidation and alloying, the proposed method, based on thermodynamic analysis and implemented in the form of freely available software InterChemBond, provides researchers with a powerful tool for preliminary screening of more than 100 thousand possible combinations of metals, alloys and oxides. This significantly accelerates the process of developing new materials, reducing the cost of experiments and minimizing the empirical approach. The paper needs improvement:
1. Lines 21–33: The introduction emphasizes the importance of interface bonds in metal oxide systems, but does not specify which specific technological applications suffer from the lack of predictive methods. Examples should be given (e.g., catalyst failure or degradation of compounds in electronics).
2. Lines 34–42: The author states that the previously developed method did not take interface reactions into account. However, it is further mentioned that it was used for multivalent oxides. A clear distinction should be made between the previous limitations and the new contribution of the present paper.
3. Lines 47–54: A method for taking interface reactions into account is claimed for the first time. However, it is not specified which types of reactions are possible from the point of view of phase diagrams and chemical potential.
4. Lines 89–103: The calculation algorithm for alloys is somewhat simplified compared to the method for pure metals. The physical basis for such a simplified scheme and its limitations in the case of highly alloyed materials should be described in more detail.
5. Lines 137–144: It is stated that in case of monovalent oxides only mixed oxides are possible. However, this generalization may be incorrect, especially for non-stoichiometric surfaces. It is advisable to discuss the limitations.
6. Lines 177–189: The entire logic for calculating the interfaces is presented as a conditional flow, but no manual calculation example is shown. It would be useful to provide a step-by-step example so that the reader can verify the correctness of the algorithm without using software.
7. Lines 218–221: The dependence of the result on the user's knowledge of the presence of a mixed oxide makes the method subjective. Automatic access to phase diagram databases (e.g. FactSage, Thermo-Calc) should be considered.
8. The paper does not consider heterophase interfaces such as metal–oxide–metal, which often occur in multilayer structures. Although they are beyond the scope of the current model, it is worth noting this as a limitation of the method.
9. The goal of the work - to predict chemical binding taking into account interface reactions - is formally achieved, however, in the section with conclusions (pp. 290-309) there is no quantitative assessment of the accuracy of the predictions
Reviewer 3 Report
Comments and Suggestions for Authors
This is an insightful paper and a solid extension of earlier work on metal–oxide interfaces. The idea of including interfacial reactions adds depth, and the tool developed is clearly useful. Still, there are a few points that could help improve clarity and usability:
1. In Table 4, some oxides that are mentioned as supported seem to be missing data. It would be good to fill in the gaps, especially for the lesser-known compounds.
2. Tables 1–3 mix theoretical predictions with experimental data, but it’s not always obvious which is which. A clearer distinction would help readers judge the model's reliability.
3. Formula (1) is explained well, but a worked-out example using numbers would make it easier to follow for those not deeply familiar with thermodynamics.
4. When it comes to using InterChemBond, it would help to mention how the software reacts if users make a wrong choice, like assuming no mixed oxide when one actually exists.
5. The flowcharts are useful, but adding a sample with actual values (like X1 and Y1) would make them more beginner-friendly.
6. Some entries in the adsorption tables show “<0” or “–” — it’s unclear what these mean. Do they stand for unknown values, negative numbers, or just skipped data? Clarifying this would help.
7. A few experimental comparisons are included, but the number is small. Maybe it’s worth pointing that out in the conclusion and suggesting that more validation is needed.
8. When there’s a mixed oxide, the model always predicts oxygen-terminated bonding. While that makes sense, noting it as a built-in limitation would be honest and helpful.
9. A short paragraph on what the method can’t yet do, or what could be improved next, would give the paper a stronger ending.
Round 2
Reviewer 2 Report
Comments and Suggestions for Authors
Overall, the authors have satisfactorily revised the paper in response to my comments. The main suggestions were implemented correctly, the structure of the paper was improved, explanations, examples and limitations were added. However, important components such as quantitative verification of accuracy, increasing objectivity (input of phase data), expanding visual explanations for new formulas require additional attention.
Add a tabular analysis of the method's prediction accuracy based on known experimental data - not necessarily a full quantitative assessment, but at least generalized statistics.
Focus in the conclusion on the limitation of the method due to the subjectivity of the input (the user chooses the presence of mixed phases).
Add visual flowcharts and brief explanations of the application of formulas (1) and (2) on several different systems.
